

**Under a new light: validation of eddy covariance flux with light response functions of**
**assimilation and estimates of heterotrophic soil respiration.**
**Georgia R. Koerber[1], Wayne, S. Meyer[1], Qiaoqi SUN[1], Peter Cale[2], Cacilia M. Ewenz[3,4]**
[1]Department of Ecology and Environmental Science, School of Biological Sciences, The
University of Adelaide, Adelaide, SA 5005, Australia
[2]Australian Landscape Trust, Riverland, Calperum Station, PO Box 955, SA 5341, Australia
[3]CSIRO Oceans and Atmosphere Flagship, Yarralumla, ACT 2600, Australia
[4]Airborne Research Australia / Flinders University, PO Box 335, SA 5106, Australia
*Correspondence to:* G.R. Koerber (georgia.koerber@adelaide.edu.au)
Keywords: ecosystem respiration, light response, autotrophic respiration, assimilation,
heterotrophic soil respiration, leaf area index, semi-arid woodland, basal soil respiration,
bushfire
Type of Paper: Primary Research Article
**Abstract.** Estimation of the basal or heterotrophic soil respiration is crucial for determination
of whether an ecosystem is emitting or sequestering carbon. A severe bushfire in January
2014 at the Calperum flux tower, operational since August 2010, provided variation in
ecosystem respiration and leaf area index as the ecosystem recovered. We propose ecosystem
respiration is a function of leaf area index and the y-intercept is an estimate of heterotrophic
soil respiration. We calculated an assimilation rate from eddy covariance data for light
response functions to calculate ecosystem respiration incorporating suppression of the
daytime autotrophic respiration. Ecosystem respiration from light response functions
correlated with data processing calculations of ecosystem respiration by OzFluxQC ($y_0$ =
$0.161x + 0.0085$; Adj. $r^2 = 0.698$). The relationship between ecosystem respiration and leaf
area index ($y_0 = 1.43x + 0.398$; Adj. $r^2 = 0.395$) was also apparent. When this approach was
compared to field measurements of soil respiration and mass balance calculations from
destructive leaf area, leaf area index calculations and litter fall, the year of data corresponding



to the year of soil respiration measurements, the y-intercept was 0.432 µmol m$^{-2}$ s$^{-1}$ or 163.44
gC m$^{-2}$ year$^{-1}$ ($y_0 = 1.37x + 0.432$, Adj. $r^2 = 0.325$). The mass balance approach for the net
primary productivity when subtracted from the tower NEE estimated heterotrophic soil
respiration of 134.59 gC m$^{-2}$ year$^{-1}$. This is only 28.9 gC different, therefore the y-intercept
approach indeed provides an estimate of heterotrophic soil respiration.

**1      Introduction**

The flux of $CO_2$ determined from eddy covariance (EC) measures and calculations is

a net value because sequestration by photosynthesising vegetation and emission from
respiration within a soil plant ecosystem occurs concurrently. The total of this flux over a day
is called the net ecosystem exchange (NEE). Partitioning EC determined NEE to quantify the
contributions of sequestration and emission of $CO_2$ is challenging. Ideally, independent
measures of both daytime and night-time ecosystem respiration are needed to make reliable
estimates of net ecosystem productivity (NEP), the amount of C retained in the ecosystem.
However, very few independent measures of respiration are made and so other methods of
estimation are used.  This paper describes one approach to improve the estimate of NEP.

At night when photosynthesis is not occurring, the flux of $CO_2$ from the ecosystem

into the atmosphere is the measure of ecosystem respiration. However the often calm night-
time atmospheric conditions are not ideal for measures of $CO_2$ flux to and from an ecosystem.
Quality EC measures depend on adequate mixing of the atmosphere moving over the soil
plant system. This is generally not an issue during daylight hours when near surface
atmospheric mixing is usually large (Burba 2013).  During the night however, air within the
vegetation layer cools and decouples from the layer of air above the plant canopy. With little
turbulent mixing, sensors at the top of the EC tower do not fully indicate near surface fluxes.
When this happens the estimate of night time $CO_2$ flux will be an unreliable measure of
ecosystem respiration (ER). To minimise the bias that this measurement limitation may
induce in full day exchange values, various data filters are applied. Most commonly,
minimum thresholds for average half hourly values of friction velocity, u[*] (Goulden et al.,
1996) are used for removing measurements when it is deemed that there is insufficient
mixing. Van Gorsel et al. (2007) identified a maximum value of $CO_2$ flux in the early evening
as the appropriate value for night-time respiration at their undulating site. This may not be an



appropriate or reliable method for flat to moderate topography sites.
In the absence of independent measures or estimates of ER, many researchers
extrapolate from night-time $CO_2$ flux into the daytime (Gilmanov et al., 2007; Lasslop et al.
2010; Wohlfhardt et al., 2005; Reichstein et al., 2005). However it is known that daylight
suppresses autotrophic respiration (Heskel et al., 2013) and hence application of night-time
derived respiration estimates during the daytime, will lead to an underestimate of plant C
sequestration, often referred to as net primary production (NPP). The reason why foliar
autotrophic respiration in the light is suppressed compared to respiration in the night is not
completely understood (Ayub et al., 2014). Not with standing this, developing improved
methods of estimating ER is important. To assist with this it is important to recognise that ER
is deemed to be the sum of two components – heterotrophic (HR) and autotrophic respiration
(AR). Autotrophic respiration is the efflux of $CO_2$ emanating from otherwise
photosynthesising organisms that fix C while heterotrophic respiration is the efflux of $CO_2$
from all organisms that derive the C from other sources. Soil respiration has an efflux of $CO_2$
from living plant roots (autotrophic) and from a plethora of soil organisms (heterotrophic)
occurring concurrently. Kuzyakov and Larionova (2005) concluded that the main reasons
why NEE and NEP are often not equal is that the C input into the rhizosphere (part of the
below ground carbon (BGC) when estimating NPP) is ignored and often there is no
accounting or limited accounting of HR.
Estimates of HR have been made from extrapolation of linear regressions between
total soil respiration and root biomass back to the y-intercept value i.e. at zero root mass
(Koerber et al., 2010; Kuzyakov, 2006; Kucera and Kirkham, 1971). The causal link between
LAI and ER is self-evident (Xu et al., 2004; Lindroth et al., 2008; Cleverly et al., 2013) and
we propose that another method of deriving an estimate of HR is to extrapolate this function
to the y-intercept with LAI = 0. However to quantify this relationship there needs to be
variation in both ecosystem respiration and leaf area index, preferably with a wide range of
both values to establish a robust relationship. In this study a wide range of LAI resulted from
measurements made before and after the woodland ecosystem was burned in a bushfire.
The procedure used in this study to develop an improved estimate of ER and in turn
HR was as follows. Daylight $CO_2$ flux values from half hourly EC measures were adjusted by
subtracting an estimate of ER derived from the immediately preceding night-time flux values.
The ratios of associated daytime to night-time soil temperature and daytime to night-time soil
water content were used to scale the expected increased respiration as daytime temperatures
were generally greater and soil water contents slightly lower than those at night. The adjusted



daytime $CO_2$ flux was an estimate of ecosystem assimilation (A, $\mu$mol m-2 s-1). Then the
relationship between A and photosynthetically active radiation (PAR, $\mu$mol m-2 s-1), more
generally known as light response functions (Cleverly et al., 2013; Wohlfahrt et al., 2005;
Lasslop et al., 2010), for each month was plotted. The use of A instead of NEE (Gilmanov et
al., 2007; Lasslop et al. 2010; Wohlfhardt et al., 2005) in the light response function is an
attempt to account for suppression of the daytime AR by light (Heskel et al., 2013; Kok,
1949; Kok, 1956). The relationship between A and PAR fits a rectangular hyperbola function
that then enables extrapolation to PAR = 0 and hence an estimate of night-time ER.

The research null hypotheses of this paper are: (i) Ecosystem respiration will not be a

function of leaf area index. (ii) Direct night-time respiration and respiration in the night
derived from light response functions (using daytime data) will not correlate with each other
and (iii) HR from NEE + NPP will not agree with HR estimated as a y-intercept from ER
versus LAI.

**2       Materials and methods**

**2.1      Site description and tower instrumentation**

The flux monitoring  site was a semi-arid mallee woodland on Calperum Station

approximately 20 km from Renmark in South Australia (34˚00.163S, 140˚35.261E; Fluxnet
site abbreviation: AU-Cpr). A 20m high EC tower, as part of the OzFlux Terrestrial
Ecosystem Research Network (TERN) was erected in June 2010 (Flight Bros. Adelaide SA)
and measurements began August 2010. The surrounding mallee ecosystem (Noble and
Bradstock, 1989) is typical of semi-arid ecosystems, adapted to long term annual median
rainfall (242 mm) encompassing drought years (Meyer, et al., 2015) and survives by
accessing occasionally replenished water stores deep in the soil profile (Mitchell et al., 2009).
The characteristic sand hills of the region run west to east with rolling undulations from
swale to crest of 5 to 8 m. The area has the largest (>1 million hectares), continuous remnant
of mallee habitat in Australia (Nulsen et al., 1986). Mallee surrounds the tower at least 10 km
in every direction. The sand hills are stabilized by eucalypt species (*Eucalyptus Dumosa,*
*Eucalyptus incrassata, Eucalyptus oleosa* and *Eucalyptus socialis*) with sparse plants of
*Eremophila, Hakea, Olearia, Senna* and *Melaleuca* genera in the mid-storey and *Triodia spp.*
in the understory.



The mean air temperature is 25 ˚C (data accessed from http: //www.bom.gov.au/) with
hot summers including days with maximum temperatures greater than 40 ˚C. The area often
experiences significant summer rainfall events of 20-60 mm in November to March after
lengthy dry periods during the year. Soils are alkaline sand (94% sand, 4% silt and 2% clay)
with an Australian classification of Tenosol (Isbell, 2002) and US Soil Taxonomy
classification of Aridisol (Soil Survey Staff, 1996). Total organic carbon, nitrogen and
carbonate (0-300 mm) are 0.5%, 0.04% and 0.25% respectively. Additional site detail and
soil properties are given in Sun et al., (submitted) and Sun et al., (2015).
The site experienced a bushfire during 15 to 19 January 2014 burning 52 713 ha with
a perimeter of 140 km according to the Country Fire Service, South Australia. The majority
of instruments on the OzFlux tower were destroyed by the fire.  These were restored within
three months to monitor ecosystem recovery. A detailed description of the EC and ancillary
instrumentation is in Meyer et al. (2015). Briefly, measurements of three-dimensional wind
speed (CSAT3 sonic anemometer, Campbell Scientific Inc., Logan, UT, USA), virtual
temperature (CSAT3), water vapour density in air and $CO_2$ density in air using an open-path
IRGA (Licor LI7500, LiCor Biosciences, Lincoln, NE, USA), were recorded at a frequency
of 10 Hz.
Auxiliary observations of solar irradiance (Es), air temperature, vapour pressure
deficit (D) and rainfall, soil temperature and soil water content were also collected
concurrently. Incident Es was observed from a four component radiometer that was
positioned at a height of 20 m (CNR4, Kipp and Zonen, Delft, the Netherlands). D was
determined as the difference between atmospheric vapour pressure (kPa) and saturation
vapour pressure at air temperature (HMP45C, Vaisala, Helsinki, Finland) at a height of 2 m.
An additional pyranometer (Licor LI2003S, LiCor Biosciences, Lincoln, NE, USA) was
mounted at 20 m and cup anemometers and wind direction sensors (RM Young, Traverse
City MI, USA) at 2 and 8.6 m. Onsite rainfall (CS7000, Hydrologic services, Warwick,
NSW, Australia) was measured with the tipping bucket gauge (0.2 mm resolution) mounted
on a stand of height 0.65 m in a clear area 8 m from the tower. Soil temperature and water
content sensors (CS650, Campbell Scientific, Townsville, Australia) were buried 10 metres
away from the tower base with multiple depths, ranging from 0.1 m to 1.8 m. Sensors were
placed in bare soil (inter-canopy) or beneath eucalypt canopies (under canopy). The collars
for measuring soil respiration in burnt Mallee were within 200 m from the tower base.
Covariances were computed every 30 min to generate fluxes following standard data
processing and quality assurance and correction procedures (Isaac et al., (In preparation for



this Special Issue); Cleverly et al., 2013; Eamus et al., 2013), hereafter referred to as
OzFluxQC. A friction coefficient ($u^*$) threshold was then calculated and set to 0.26 m s$^{-1}$,
0.21 m s$^{-1}$, 0.23 m s$^{-1}$, 0.25 m s$^{-1}$, 0.26 m s$^{-1}$ and 0.26 m s$^{-1}$ for the years 2010, 2011, 2012,
2013, 2014 and 2015 respectively.

To calculate the effective sampling footprint of the tower we used the Kormann-

Meixner method (Kormann and Meixner, 2001), employing a modified version of the ART
Footprint Tool of Neftel et al. (2008). The Kormann-Meixner footprint determines the two-
dimensional density function for an ellipse upwind from the tower. The predominant wind
direction here is from the south-westerly quarter. For every 30 minute measurement of wind
speed and direction, mixing and buoyancy parameters the data is filtered according to the
Kormann-Meixner constraints. Analysis of the seasonal effects exhibited a smaller footprint
in summer which reflected the increased mixing in summer as well as the influence of more
frequent winds from the northerly quarter. The annual average of the footprint area for 2014
displayed a distance from the tower of 500 m for at least 10% of the maximum contribution
(1300 m for at least 1%).

The regression of latent energy plus sensible heat (LE + H) against net radiation plus

soil heat flux (Rn + G) was used to check energy balance closure. From 1 August 2010 to 31
August 2013 the relationship was (LE + H) = 0.8769 (R$_n$ + G) + 2.5095, r$^2$ = 0.9159. This
indicated that energy balance was not completely achieved, as is commonly observed with
the eddy covariance method (Twine et al., 2000).

**2.2    Light response functions**

The light response function needed was the relationship between the assimilation rate

(A) and the incoming radiant energy. Assimilation was partitioned from NEE as shown in the
schematic flow chart (Fig. 1). To calculate A from NEE the daytime values of NEE were
increased in absolute magnitude by the expected rate of $CO_2$ emission from the soil and plant
system. The daily night-time 30 minute respiration (AR + HR) values were adjusted using the
ratio of average daytime soil temperature to the night-time soil temperature. A further,
generally minor adjustment was made using the ratio of average daytime to night-time soil
water content measured at 100 mm depth. The adjusted night-time average value was then
subtracted from each daytime 30 minute flux to give an assimilation (A) rate with an absolute
value greater than NEE. The calculation of A for every 30 minutes of the daytime in each
month was then regressed against short wave radiant energy converted to photosynthetically



active radiation (PAR) in µmol m$^{-2}$ s$^{-1}$ according to Meek et al., (1984) and McCree (1972) as
detailed in Biggs (1984).
A rectangular hyperbola was fitted to the 30 minute data each month (Eqn. 1,
Wohlfahrt et al., 2005; Lasslop et al., 2010; Cleverly et al., 2013) with starting values of -10,
300 and 0.5 for the net saturated A (V$_{max}$), saturating PAR (K$_m$) and constant (c) respectively,
all in µmol m$^{-2}$ s$^{-1}$. The value of A when PAR = 0 was assigned as the night-time respiration
(R$_{night}$) value for that month. Further, rearranging the same equation and solving for the value
of PAR when A = 0 (Eqn. 2) gave the compensation point when low PAR and hence
photosynthesis no longer compensated respiration (Heskel et al., 2013). When PAR was
greater than this compensation point, ER was deemed to be supressed by the incoming
radiant energy.

$A = V_{max} \times (PAR / (K_m + PAR)) + c$        Eqn. 1

Where V$_{max}$ is the light saturated net photosynthetic rate

K$_m$ is the saturation light intensity

c is a constant


$PAR = (K_m (A - c)) / (V_{max} - A + c)$      A = 0      Eqn. 2

Fitting the rectangular hyperbola model used the SPSS procedure (IBM SPSS Statistics V. 21
New York, US) of nonlinear weighted least squares fitting using the Levenberg-Marquardt
algorithm.

2.3    Leaf area index
During May 2013 to September 2015, plant area index (PAI) of the canopy above 0.5 m from
the ground was measured optically using the digital cover photography method (DCP) (Pekin
and Macfarlane, 2009, Macfarlane et al., 2007) as described in Eamus et al., (2013). A 1 ha
(100 m x 100 m) area immediately to the north west of the tower was marked and 10 x 100 m
transects were identified along which photographs were taken at 10 m intervals. Photographs
were taken using a Sony Nex-7 DSLR camera fitted with a lens of 25 mm focal length. The
camera settings were automatic exposure, aperture-priority mode, F-stop of 9.0 and ISO 400.
The camera was oriented to 0° nadir (viewing upward). Calculation of PAI used an extinction



coefficient of 0.5. For eight months after the fire the photographs taken were of the trunks
and branches without leaves. This area could be subtracted from the previously determined
plant area to obtain LAI.
For cross calibration purposes leaf area was determined directly by destructively
collecting epicormic stem and leaf regrowth of five trees in April 2015, approximately one
year after the bushfire. Leaves from a stem were removed, and a subsample of leaves was
measured with a leaf area meter. The subsample and main leaf sample were weighed after
oven drying at 60˚C for 48 hours, and the specific leaf area of the subsample was used to
calculate the whole tree leaf area.

**2.4     Soil respiration, litter collection, tree spacing and biomass**
Soil $CO_2$ efflux was measured monthly from July 2014 to June 2015 (total 12 sampling
campaigns) with a manual chamber connected to an infra-red gas analyser (LI-8100, LI-COR
Inc., Lincoln, Nebraska, USA). Details are in Sun et al., (accepted May 2016).
In May 2013, 3 litter trays ($450 \times 340 \times 55$ mm aluminium BBQ trays) were placed in
the 1 ha area adjacent to the tower. These were dug in and secured so that the upper edge was
flush with the ground surface. Litter was collected monthly, dried at 60 ˚C for 48 hours and
weighed. The carbon content was assumed to be 35% of plant material dry mass (Hadley and
Causton, 1984).
On 17 June 2014 remnant (burnt) tree trunks within the 1 ha area adjacent to the tower
were viewed aerially, without the obstruction of any leaf canopy using a 3D Robotics RTF
Y6 conservation drone. Images were captured at 70 metres above ground at a resolution of
21.6 mm per pixel in RGB colour. Images were mosaicked with Pix4Dmapper and improved
by referencing to an existing ortho-rectified aerial photographic image. The central point of
each mallee tree was marked with a digital dot while viewing the imagery at scale of 1:100 in
ArcGIS. The mean distance between trees could then be calculated and this spacing used to
scale up biomass and LAI from the sub sample measurements.
The total carbon associated with the 1 ha area was estimated from the measurements
of tree numbers and dry mass of eight destructively sampled trees. This enabled an estimate
of aboveground carbon (AGC). An estimate of belowground carbon (BGC) was made using
soil respiration measurements and litter amounts (Koerber et al., 2009; Clark et al., 2001;
Raich and Nadellhoffer, 1989; Nadellhoffer et al., 1998).



## 3 Results

The results were determined primarily from the light response functions and the extrapolated values of respiration in the night from daytime A. These values reflect the environmental conditions the mallee ecosystem was experiencing each month of a year.

### 3.1 Net ecosystem exchange

During the four years prior to 2010, the annual average rainfall was 215 mm, with each year being consistently below the long term median annual rainfall of 242 mm. These dry years were part of a prolonged dry period generally referred to as the "Millennium drought". Significant rain (259 mm) fell in the last five months of 2010, the Millennium drought ended and the mallee ecosystem became a C sink with monthly NEE of -15.49 g C $m^{-2}$ $month^{-1}$ for December 2010. During 2011, with further rain (511 mm for the year) the mallee responded and recovered as indicated by an increase in NEE to -25.70 g C $m^{-2}$ $month^{-1}$ for July 2011 and a maximum of –44.46 g C $m^{-2}$ $month^{-1}$ in April 2011. This increased uptake of C corresponded to an observed increase in green leaf canopy of both trees and grass cover that was reflected in increased remotely sensed NDVI values and inferred LAI back calculated from latent energy exchange determined by the EC measurements (Meyer et al. 2015). This response is consistent with the wide area response during March to May 2011 of Australian arid and semi-arid vegetation to the summer rainfall of 2010 – 2011 (Poulter et al., 2014; Cleverly et al., 2016). During 2012, the recovered ecosystem was sustained during the first half of the year with maximum NEE of -42.83 g C $m^{-2}$ $month^{-1}$ in April 2012. The second half of 2012 was dry (62 mm of rain) and this lower than average rainfall continued into most of 2013. In 2013 the maximum NEE was only -17.82 g C $m^{-2}$ $month^{-1}$ in August. This rate is similar to that recorded at the end of the Millennium drought in late 2010. In January 2014 the destruction of the vegetation in the bushfire resulted in the ecosystem becoming a carbon source, with a maximum emission of 13.53 g C $m^{-2}$ $month^{-1}$ recorded in May 2014. Signs of vegetation recovery were evident in July 2014 as the mallee trees sprouted epicormic stems and juvenile leaves from the lignotubers. In the months of August and September 2014, NEE was -7.73 and -7.59 g C $m^{-2}$ $month^{-1}$ respectively. In 2015, the ecosystem was a sink with a maximum NEE of -20.75 g C $m^{-2}$ $month^{-1}$ in June. Annual NEE from OzFluxQC for each year along with the partitioning into gross primary productivity (GPP) and ER are given in Table 1.




### 3.2    Assimilation light response functions


The half hourly assimilation (A) values and associated radiation (PAR) values for each month
of the entire measurement period were plotted and the assimilation light response function
fitted (Table 2). In the summer of 2012, throughout 2013, and the spring and summer of
2015, when the mallee ecosystem was dry, regression $r^2$ were higher with PAR threshold <
1500 µmol m$^{-2}$ s$^{-1}$. Even so the regressions had higher coefficients during the winter months
and were lower in summer months. This likely indicates that assimilation was more
constrained by available radiation in the cooler, less evaporative winter months, while in
summer, assimilation was constrained by greater stomatal control as water availability to
meet high evaporative demand was limiting (Ayub et al., 2011; Meyer et al., 2015).
The relationship between night-time respiration, derived from the flux tower
measurements using OzFluxQC processing against night-time respiration determined
indirectly from the y-intercept of daytime A and PAR response functions (Fig. 2) are
significantly correlated and approximately similar in the years preceding the bushfire
although 2013 was experiencing drought (Pearson correlations, 2010: $r = 0.873$, $P \leq 0.05$;
2011: $r = 0.58$, $P \leq 0.05$; 2012: $r = 0.615$, $P \leq 0.05$; 2013: $r = 0.27$, $P = 0.396$, Fig. 2). In 2014
after the bushfire, all values were small ($< 0.7$ µmol m$^{-2}$ s$^{-1}$) with the flux tower values
generally being larger than those derived from the light response functions. In 2015, night-
time respiration from the tower and from light response curves continued to be small. The
spread of respiration values determined from the assimilation light response function is
similar in 2014 and 2015 but was smaller than those estimated in the years before the
bushfire.

### 3.3    Comparison of ER from (NEE – A) and ER from OzFluxQC

Calculation of ER as (NEE – A) was significantly correlated to ER from the
processing by OzFluxQC, Pearson $r = 0.838$ $P \leq 0.0001$ (Fig. 3). From the equation of the
line ($y_0 = 0.1612x + 0.0085$, $r^2 = 0.6977$), the OzFluxQC is underestimating ER with smaller
positive rates compared to ER from a calculated A. The larger positive ER corresponds to a
more negative ER if using the convention of negative rates for respiration (Atkin et al., 2013)
and is in line with their statements that not incorporating supressed daytime respiration
underestimates ER.




### 3.4 Relationship between ER and LAI and estimates of HR

The relationship between ER derived from (NEE − A) and LAI for 25 months around the
bushfire was highly significant (Fig. 4; $y_0 = 1.43x + 0.398$; Adj. $r^2 = 0.395$, Pearson
correlation, $r = 0.648$ $P \le 0.0001$). From this relation the inferred ER for this period is 0.398
$\mu$mol m$^{-2}$ s$^{-1}$. The ecosystem respiration was standardized to 20 ˚C and 0.03 g g$^{-1}$ soil water
content to remove seasonal variation. There are three outlier points with apparently
suppressed ER for the months of April, May and June 2014, immediately after the bushfire.
For the period from July 2014 to June 2015 that corresponds to the year that in-situ soil
respiration measurements were made post fire, the y-intercept is 0.4316 $\mu$mol m$^{-2}$ s$^{-1}$ ($y_0 =$
$1.365x + 0.4316$, Adj. $r^2 = 0.3249$, Pearson correlation $r = 0.570$ $P = 0.053$). The value at LAI
$= 0$ gave an estimate of ER and more particularly HR of 163.44 gC m$^{-2}$ year$^{-1}$.
An alternative approach to estimate HR is to calculate the sum of AGC and BGC, that
is effectively net primary production (NPP), and subtract OzFluxQC derived NEE. Using the
mean ground area per tree of 16 m$^2$ derived from drone imagery, the annual increase in AGC
was estimated to be $105.68 \pm 27.37$ gC m$^{-2}$ year$^{-1}$. For July 2014 to June 2015, soil
respiration was estimated to be 490.72 gC m$^{-2}$ (details in Sun et al., 2016), litter fall was
$566.17 \pm 62.57$ gC m$^{-2}$ and hence BCG was 75.45 gC m$^{-2}$ year$^{-1}$. The sum of AGC and BGC
and therefore NPP is 181.13 gC m$^{-2}$ year$^{-1}$. With NEE for the year of $-46.54 \pm$ gC m$^{-2}$ year$^{-1}$
the estimate of HR is 134.59 gC m$^{-2}$. This compares very favourably with the estimate
(163.44 gC m$^{-2}$) from light response functions and is 44% of NEE. This coincidence indicates
that the method of extrapolation of the assimilation (A) and incoming energy (PAR)
relationship to PAR = 0 (i.e. the y-intercept) provides an estimate of ER each month
incorporating AR.

### 4 Discussion


In this paper we have demonstrated another way to partition NEE recorded by EC
towers into the C sequestered by photosynthesis and the efflux of C from respiration.
Calculation of daily NEP using an estimate of ER from extrapolation of ecosystem light
response functions using A instead of NEE, indicates that derived NEP is inevitably larger
i.e. the NEE light response function usually overestimates daily respiration (Ayub et al.,



2011; Heskel et al., 2013). The method of estimating HR from the extrapolation of the ER
(NEE-A) versus LAI, is similar to that of estimating HR from the y-intercept of soil
respiration and root mass (Koerber et al., 2010; Kuzyakov, 2006; Kucera and Kirkham,
1971). The concept of the y-intercept providing an estimate of heterotrophic soil respiration
from the assimilation light response function is novel and hasn't been used to assist
partitioning EC derived NEE.

The estimates for HR of 163.44 gC m$^{-2}$ year$^{-1}$ from light response function derived ER

versus LAI or 134.59 gC m$^{-2}$ year$^{-1}$ from (NEE + NPP) are equivalent to 1.63 tC ha$^{-1}$ year$^{-1}$
and 1.34 tC ha$^{-1}$ year$^{-1}$ respectively. As expected, these are lower but of the same order of
magnitude as that estimated (8.13 tC ha$^{-1}$ year$^{-1}$) in much wetter and more plant productive
vegetable farming regions in the UK (Koerber et al., 2009).

Partitioning of NEE derived from EC measurements indicates that in semi-arid

environments, the timing of rainfall relative to preceding drying greatly influences the
outcome of the dynamic balance between sequestration and respiration. For example, Xu et
al. (2004) found that in a Mediterranean grassland the early onset of rain in the winter
growing season resulted in C assimilation i.e. gross primary productivity (GPP) to be greater
than ER and NEE was negative i.e. the ecosystem was a carbon sink. However if significant
rainfall did not occur until late in spring or early summer and the water stressed grass was
dead, ER was greater than GPP and NEE was positive i.e. the ecosystem was a carbon source.
Monthly values of NEP and ER derived in this study suggest that the timing of rainfall in
relation to the preceding dry or wet period was more important in determining the net C
balance of the ecosystem than the total amount over the course of a year. Paul Jarvis's
research (Jarvis et al., 2007) on soil respiration pulses after rain, carrying on the discovery by
H.F. Birch 50 years ago (the "Birch" effect) showed the same effect. His research and that of
Xin Wang et al (2014) suggests that increased rainfall in summer, along with increasing
ambient temperature from global warming will increase the contribution of HR in soil
respiration. Soil respiration pulses following rainfall may be enhanced by the availability of
organic breakdown materials coming from photo-degradation during drought periods (Ma et
al., 2012). Rainfall that irregularly occurs in persistently arid areas such as the *Corymbia*
savanna and Mulga ecosystems of inland Australia seems to cause net carbon loss at least in
the short term (Cleverly et al., 2016).

The relationship between direct and indirect derived night-time respiration shown in

Fig. 2 was close to 1:1 during 2010 and 2011. Drying in 2012 persisted into 2013 and this
seems to have affected this relationship. With the bushfire in 2014 there was no active



photosynthetic canopy and only a small but increasing amount in 2015, the amount of
respiration declined presumably because both AR and HR declined – AR because the
majority of the above ground growth was dead and HR because there is no supply or little
supply of photosynthetically derived C from the above ground system to below ground. The
reasons why EC estimates of night-time respiration in 2014 appear to be large relative to the
light response function is uncertain. With loss of the tree, mid story and ground canopy the
atmospheric exchange and mixing would be different. It is not clear why this may cause what
appears to be an over-estimate of the $CO_2$ flux. However it is equally possible that the light
response functions are underestimating the flux since active leaf area is very low and hence
assimilation is very limited.

After careful consideration, two more problems had to be reconciled. The first is

calculations of assimilation were an underestimate in the outset. With ER equilibrated in the
night and the day from a ratio of the soil temperature and soil water content in the night and
the day, subsequent subtraction covers over some of the A seen as respiration in the day is in
fact suppressed (Heskel et al., 2013). For example if ER was constant in the night and the day
at 3 $\mu mol\ m^{-2}s^{-1}$ and the photosynthetic rate is  -8 $\mu mol\ m^{-2}s^{-1}$, when the night is subtracted
away from the day we are left with an assimilation of -2 $\mu mol\ m^{-2}s^{-1}$ however if ER is 1 $\mu mol$
$m^{-2}s^{-1}$ in the day then assimilation will be -4 $\mu mol\ m^{-2}s^{-1}$. Therefore we had an assimilation
rate that was an underestimate. In the future we aim to develop methods for conducting linear
regressions to estimate autotrophic respiration in the daylight (Heskel et al., 2013; Kok, 1949;
Kok, 1956) for correcting the underestimate of A (Koerber et al., unpublished).

The second problem is whether our calculations of A require correction like *in vivo*

construction of light response functions requiring A versus $CO_2$ partial pressure ($p_i$) curves at
three low light intensities (Villar et al., 1994; Kirschbaum and Farquhar, 1987). As $p_i$ is
increased at low light intensity, measurements of A increase. Therefore $p_i$ should be
standardized for all light intensities and A adjusted to ensure foliar AR provides a correct
estimate of the Kok effect and hence A is not an overestimate. Our tower measurements
provided multiple A estimates at each light intensity with an external $CO_2$ partial pressure
that was reasonably constant. With this setting the EC derived light response functions do not
require standardization.

**5        Conclusion**

The advantage of using the light response function approach to determine respiration



when PAR = 0 is that it is non-destructive. The ecosystem remains intact, soils are not
disturbed and there is no need to measure respiration of the plants directly with all of the
attendant problems of sufficient sampling to assure representativeness. In this study we did
field measurements that were destructive but only to the extent of AGC necessary for the
NPP. The BGC was estimated from litter collections and soil respiration. This study
highlights the importance of measuring soil respiration as an adjunct measurement.
The similarity in heterotrophic soil respiration estimated by field measurements and
from the determination of assimilation from partitioning the NEE as described here is
encouraging, only 28.85 gC m$^{-2}$ difference. This result indicates that the NEE and NEP are
balanced at our site and we did not underestimate NEP from our field measurements. From
our initial calculations, our measurements provide rarely available evidence of the large
contribution of basal soil respiration (44%) to the total C balance. Management of the land by
land use managers needs to minimize the formation of ecosystems susceptible to larger
emissions of basal soil respiration arising from our changing climate. There is much to gain
from understanding dry and arid ecosystem functioning of the plants within the sandy
alkaline soils of southern Australia. Mallee's are an important biomass crop, potentially
providing an increasing income from payments for carbon sequestration, for landholders.
This study has been able to reject all three null hypotheses. When the hypotheses are
addressed in reverse order, firstly, we were able to estimate the heterotrophic soil respiration
from field measurements and the y-intercept of ecosystem respiration versus leaf area index.
Secondly, light use efficiency functions for the respiration in the dark from rectangular
hyperbola agree with direct night time data. Lastly, ecosystem respiration is a function of
LAI.












460 **Table 1**. Annual GPP, ER, NEE in gC m$^{-2}$ year$^{-1}$ and rainfall for 2011 to 2015. Values are

461 from OzFluxQC. Measurements started at the tower in August 2010 and GPP, ER, NEE and

462 rainfall are sums for August to December 2010 (5 months).


| Year | GPP | ER | NEE | Rainfall mm |
|------|--------|--------|---------|-------------|
| 2010 | 100.74 | 29.84  | -70.9   | 259.0       |
| 2011 | 432.05 | 114.37 | -317.68 | 510.8       |
| 2012 | 377.84 | 93.68  | -284.16 | 211.2       |
| 2013 | 237.15 | 68.73  | -168.41 | 242.4       |
| 2014 | 52.03  | 32.57  | -19.46  | 211.6       |
| 2015 | 155.02 | 56.55  | -98.47  | 241.4       |




**Table 2.** Coefficients from assimilation light response functions. Units are µmol m$^{-2}$ s$^{-1}$.
Rainfall in brackets is from Renmark when the EC measurement system was not in operation
after bushfire.

| | Rainfall mm | Compensation point when Fc = 0 | Vmax | Km | ER in **night**, $r^2$ and $n$ from rectangular hyperbola |
|---|---|---|---|---|---|
| | | Units are all µmol m$^{-2}$ s$^{-1}$ | | | |
| **2010** | | | | | |
| July | 6.0 | 76.3 | -3.2 | 456.8 | 0.47 $r^2$=0.74 $n$=25 |
| August | 24.0 | 96.5 | -3.9 | 392.8 | 0.78 $r^2$=0.41 $n$=520 |
| September | 48.0 | 110.2 | -4.5 | 558.3 | 0.74 $r^2$=0.40 $n$=565 |
| October | 67.0 | 140.8 | -4.7 | 825.5 | 0.68 $r^2$=0.26 $n$=633 |
| November | 35.0 | 160.4 | -4.4 | 331.4 | 1.44 $r^2$=0.28 $n$=615 |
| December | 86.0 | 115.3 | -4.8 | 316.5 | 1.28 $r^2$=0.25 $n$=749 |
| **2011** | | | | | |
| January | 87.4 | 135.5 | -7.3 | 577.9 | 1.39 $r^2$=0.36 $n$=743 |
| February | 109.0 | 153.4 | -9.9 | 522.9 | 2.24 $r^2$=0.46 $n$=604 |
| March | 63.4 | 114.7 | -12.1 | 887.6 | 1.39 $r^2$=0.61 $n$=613 |
| April | 4.4 | 73.3 | -12.1 | 1024.0 | 0.81 $r^2$=0.52 $n$=541 |
| May | 14.2 | 86.4 | -11.2 | 576.3 | 1.46 $r^2$=0.52 $n$=445 |
| June | 4.8 | 72.7 | -9.7 | 647.3 | 0.98 $r^2$=0.61 $n$=459 |
| July | 13.4 | 85.7 | -7.8 | 493.2 | 1.15 $r^2$=0.56 $n$=497 |
| August | 25.0 | 102.0 | -7.6 | 492.7 | 1.30 $r^2$=0.38 $n$=510 |
| September | 8.2 | ..108.9 | -5.7 | 460.5 | 1.09 $r^2$=0.28 $n$=577 |
| October | 29.8 | 99.4 | -5.0 | 316.8 | 1.20 $r^2$=0.26 $n$=656 |
| November | 59.4 | 166.2 | -5.6 | 515.8 | 1.37 $r^2$=0.29 $n$=667 |
| December | 91.8 | 98.4 | -6.8 | 793.0 | 0.75 $r^2$=0.27 $n$=769 |
| **2012** | | | | | |
| January | 27.4 | 122.7 | -7.0 | 423.9 | 1.56 $r^2$=0.40 $n$=777 |
| February | 86.8 | 85.7 | -5.8 | 689.1 | 1.65 $r^2$= 0.24 $n$=657 |
| March | 9.4 | 150.7 | -10.4 | 215.5 | 1.86 $r^2$=0.49 $n$=643 |
| April | 5.4 | 86.7 | -9.0 | 456.9 | 1.44 $r^2$=0.36 $n$=550 |
| May | 9.0 | 86.5 | -8.8 | 380.6 | 1.64 $r^2$=0.52 $n$=503 |
| June | 11.0 | 89.3 | -10.3 | 743.7 | 1.10 $r^2$= 0.61 $n$=452 |
| July | 23.6 | 109.5 | -9.3 | 753.2 | 1.18 $r^2$=0.56 $n$=485 |
| August | 8.2 | 89.2 | -8.6 | 743.5 | 0.92 $r^2$=0.64 $n$=571 |
| September | 5.8 | 96.2 | -6.1 | 353.2 | 1.31 $r^2$=0.14 $n$=488 |
| October | 3.8 | 88.6 | -4.7 | 626.5 | 0.59 $r^2$=0.15 $n$=445 |
| November | 14.0 | 65.1 | -3.5 | 222.7 | 0.79 $r^2$=0.29 $n$=432 |
| December | 6.8 | 61.8 | -4.8 | 726.1 | 0.38 $r^2$=0.22 $n$=318 |





| 2013 | | | | | |
|---|---|---|---|---|---|
| January | 1.0 | 44.1 | -3.7 | 613.6 | 0.25 $r^2$=0.14 $n$=305 |
| February | 36.8 | 105.0 | -3.9 | 772.8 | 0.47 $r^2$=0.08 $n$=275 |
| March | 2.2 | 184.9 | -7.6 | 1065.8 | 1.12 $r^2$=0.31 n=465 |
| April | 13.6 | 48.3 | -10.5 | 1578.7 | 0.81 $r^2$=0.31 $n$=266 |
| May | 35.4 | 81.8 | -6.5 | 1792.4 | 0.61 $r^2$=0.39 $n$=208 |
| June | 32.0 | 138.3 | -9.7 | 617.5 | 1.77 $r^2$=0.63 $n$=275 |
| July | 24.4 | 88.1 | -10.8 | 1249.5 | 0.71 $r^2$=0.60 $n$=392 |
| August | 10.6 | 90.8 | -9.5 | 1045.7 | 0.78 $r^2$=0.58 $n$=353 |
| September | 26.2 | 160.4 | -6.0 | 578.3 | 1.30 $r^2$=0.30 $n$=396 |
| October | 9.2 | 60.2 | -4.0 | 350.7 | 0.59 $r^2$=0.24 $n$=221 |
| November | 3.6 | 23.6 | -3.9 | 508.1 | 0.17 $r^2$=0.14 $n$=435 |
| December | 27.2 | 46.3 | -3.8 | 339.0 | 0.46 $r^2$=0.16 $n$=303 |
| 2014 | | | | | |
| January | (6.8) | 45.8 | -5.2 | 1087.1 | 0.41 $r^2$=0.14 $n$=165 |
| February | (84.4) | | | | |
| March | (15.0) | | | | |
| April | (35.6) | -1695.8 | -0.4 | 532.1 | 0.55 $r^2$=0.00 $n$=66 |
| May | 28.4 | -0.8 | 0.7 | 39.8 | 0.02 $r^2$=0.01 $n$=334 |
| June | 13.8 | 2364.7 | -1.0 | 1371.1 | 0.65 $r^2$=0.02 $n$=368 |
| July | 4.6 | 333.3 | -1.0 | 1204.0 | 0.22 $r^2$=0.02 $n$=382 |
| August | 18.8 | 131.6 | -4.2 | 1973.6 | 0.27 $r^2$=0.23 $n$=477 |
| September | 6.6 | 89.4 | -2.8 | 1331.1 | 0.18 $r^2$=0.12 $n$=317 |
| October | 0.6 | 92.3 | -1.4 | 171.4 | 0.50 $r^2$=0.06 $n$=618 |
| November | 9.2 | 95.5 | -1.2 | 378.4 | 0.24 $r^2$=0.03 $n$=686 |
| December | 13.8 | 135.6 | -2.0 | 520.8 | 0.40 $r^2$=0.08 $n$=696 |
| 2015 | | | | | |
| January | 68.8 | 104.4 | -2.9 | 584.1 | 0.44 $r^2$=0.30 $n$=722 |
| February | 0.6 | 93.0 | -2.7 | 459.3 | 0.45 $r^2$=0.21 $n$=378 |
| March | 0.0 | 106.2 | -2.1 | 271.2 | 0.60 $r^2$=0.16 $n$=420 |
| April | 65.4 | 129.3 | -2.7 | 411.3 | 0.65 $r^2$=0.20 $n$=448 |
| May | 9.8 | 97.2 | -6.8 | 638.2 | 0.90 $r^2$=0.61 $n$=376 |
| June | 17.8 | 80.7 | -7.5 | 671.4 | 0.80 $r^2$=0.38 $n$=280 |
| July | 6.0 | 84.7 | -6.0 | 562.0 | 0.78 $r^2$=0.57 $n$=362 |
| August | 20.4 | 110.7 | -5.8 | 524.6 | 1.01 $r^2$=0.41 $n$=469 |
| September | 17.0 | 102.8 | -4.2 | 368.5 | 0.92 $r^2$=0.28 $n$=561 |
| October | 1.5 | 16.3 | -3.4 | 2480.4 | 0.02 $r^2$=0.08 $n$=232 |
| November | 23.0 | 21.0 | -2.4 | 774.2 | 0.06 $r^2$=0.10 $n$=419 |
| December | 0.0 | 91.4 | -1.6 | 661.5 | 0.19 $r^2$=0.07 $n$=419 |







≤ 10 W m$^{-2}$                                                          > 10 W m$^{-2}$

```
                              ┌─────────────────────────────┐
                              │   30 min. EC flux (NEE) data │
                              └─────────────────────────────┘
                                  ╱                        ╲
                          ╱                            ╲
        ┌──────────────────────────────┐        ┌──────────────────────────────┐
        │ Night 30 min. EC flux (NEE) data │    │ Day 30 min. EC flux (NEE) data │
        └──────────────────────────────┘        └──────────────────────────────┘
                     │         495
                 ┌────────┐
                 │ u* filter │
                 └────────┘
```






$$\frac{\text{Average day ST and SWC}}{\text{Average night ST and SWC}} \times \text{Night 30 min EC flux (NEE) data}$$


Day 30 min EC flux (NEE) data – Average adjusted Night EC flux (NEE) data

Day 30 min. A data

A versus PAR light response functions each month. y-intercept estimates respiration at night for each month

Monthly NEE – monthly A for monthly ER versus monthly LAI. y-intercept estimates heterotrophic soil respiration






**Figure 1.** Schematic flow chart showing the method for partitioning carbon fluxes.





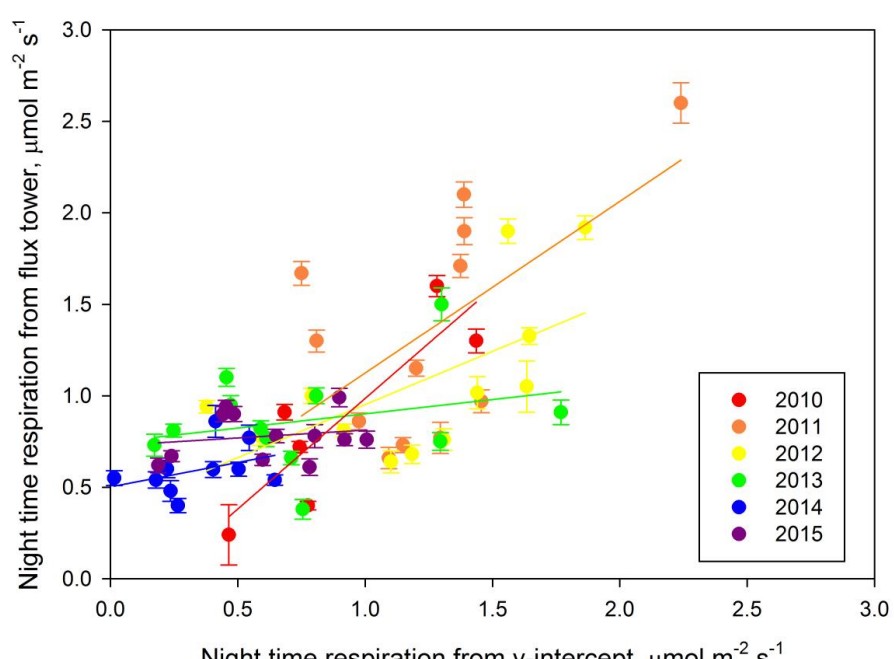

**Figure 2.** Night respiration from the EC tower measurement system and the y-intercept

approach with daytime data.

2010: $y_0 = 1.20x - 0.22$ Adj. $r^2 = 0.7617$

2011: $y_0 = 0.94x + 0.19$ Adj. $r^2 = 0.34$

2012: $y_0 = 0.58x + 0.37$ Adj. $r^2 = 0.3783$

2013: $y_0 = 0.16x + 0.75$ Adj. $r^2 = 0.07$

2014: $y_0 = 0.27x + 0.50$ Adj. $r^2 = 0.1477$

2015: $y_0 = 0.09x + 0.73$ Adj. $r^2 = 0.0312$





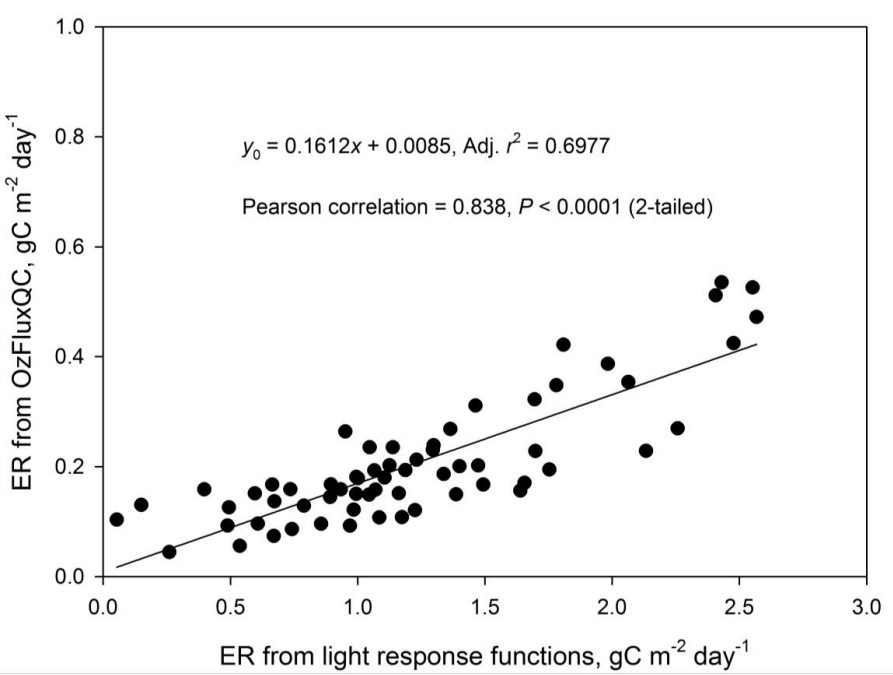


**Figure 3.** Comparison of ecosystem respiration from the OzFluxQC processing and the light

response function of calculated assimilation.






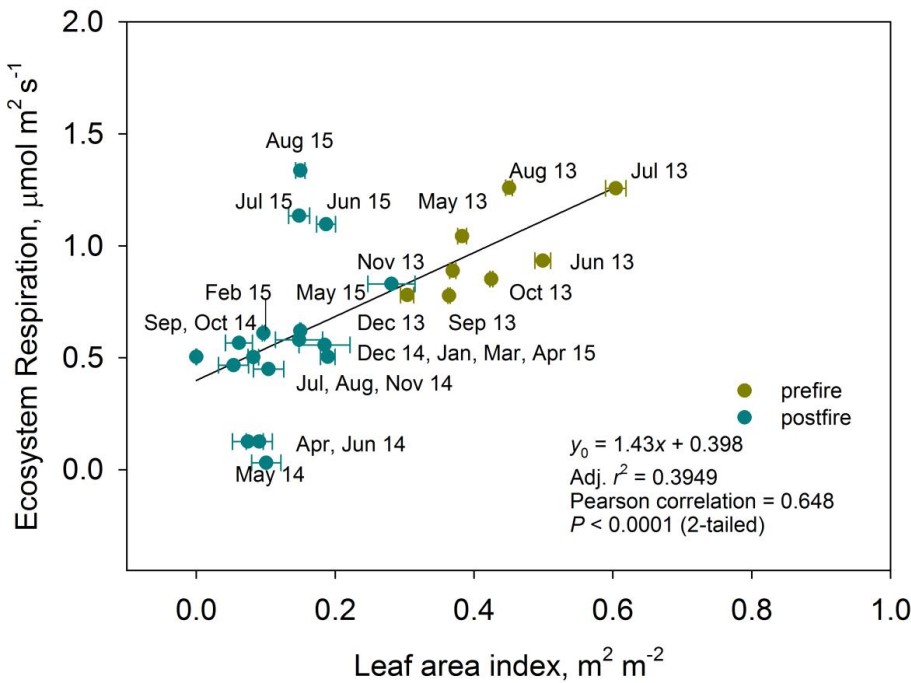


**Figure 4.** Comparison of ecosystem respiration from the light response function with

calculated assimilation extrapolated to LAI = 0 and LAI from digital cover photography. The

ecosystem respiration was standardized to 20 ˚C and 0.03 g g$^{-1}$ soil water content.
















*Author contributions.* G.R. Koerber and W.S. Meyer designed the experiment and carried it
out. G.R. Koerber, P. Cale, Q. Sun, W.S. Meyer and C. M. Ewenz performed field work.
G.R. Koerber, W.S. Meyer and C. M. Ewenz performed data collection and processing. G.R.
Koerber and W.S. Meyer prepared the manuscript with contributions from all co-authors.

*Acknowledgements.* This work was partly supported by grants from the Australian
government's Terrestrial Ecosystems Research Network (TERN) (www.tern.org.au). TERN
is a research infrastructure facility established under the National Collaborative Research
Infrastructure Strategy (NCRIS) and Education Infrastructure Fund, Super Science Initiative,
through the Department of Industry, Innovation, Science, Research, and Tertiary Education.
Thank you to 12 National Australia Bank employees under the support and expert guidance
of Cassandra Collins from the Earthwatch Institute and Peter Cale from the Riverland
Australian Landscape Trust.

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
