# Peer review of "Under a new light: validation of eddy covariance flux with light response functions of"

_Biogeosciences, 2016_

## Referee Comment (RC1) · Anonymous Referee #1 · 4 Jul 2016

The manuscript presents a promising idea for partitioning ecosystem respiration into autotrophic and heterotrophic respiration: Since leaf area index (LAI) is directly related to the autotrophic respiration, the y-intercept from the regression of ecosystem respiration against LAI would be the heterotrophic respiration. Due to a severe bush fire, the ecosystem around the eddy covariance (EC) flux tower was severely damaged and therefore LAI showed a wide variation over the six reported years to test this hypothesis.

However, there seems to be error in reasoning in the light response approach for esti-

mating the ecosystem respiration:

- The net ecosystem exchange (NEE) is first partitioned into average adjusted night EC flux and assimilation A. Hence A is is purely the assimilation WITHOUT nighttime respiration. However, in the light response approach, A is fitted against light with an offset C and for zero light, C is described and discussed as nighttime respiration $R\_night$ (which has already been subtracted beforehand). That does not seem to make any sense or needs to be clarified.

- Furthermore, the offset C is quite small for all calculated months (Table 2) with values below the random error typical for EC measurements (cf. Richardson, A.D. et al., 2008. AgForMet, 148(1): 38-50.). Hence, C might be the offset due to the random error in the measurements and would be desired as a sign that the partitioning scheme shown in Figure 1 (left flow) worked.

- Some of the monthly regressions in Table 2 yielded ecophysiological implausible values (such as negative light compensation points) and other so low r2 that it is questionable if the light response can be fitted for these months at all (since light does not seem to be the primary driver of the ecosystem response).

- Since the main result of the LAI regression again ecosystem respiration in Figure 4 is derived from monthly ER as the difference between monthly NEE and monthly A (hence no light response function used, right flow in Figure 1), it might be recommendable to totally exclude the light response approach from this manuscript.

The other major concern is the robustness of the main results in Figure 3 and in Figure 4 of "monthly NEE minus monthly A" against LAI:

- To my understanding, the used partitioning scheme is the right flow of Figure 1 and does not use the light response equation but directly calculates monthly ER from monthly NEE minus monthly A. However, it is often referred to "light response function of calculated assimilation" e.g. in Figure 3. This would need to be clarified or revised

throughout the ms.

- It is interesting that the ecosystem respiration estimates from partitioning with OzFlux are so much lower, only 16%, than the estimates found with the new partitioning scheme after Figure 1, right flow. This would definitely require further investigation and discussion.

- The main result in Figure 4 show monthly data points for prefire and postfire. The two clouds of data points have very different properties in terms of scatter as well the slope/intercept. As a quick test (by reading the values from the figure), a linear regression of only the postfire data yielded a y-intercept of 0.23 and a very low r2 value of 0.26. The y-intercept is thus only half of the regression of the postfire data points. The disturbed ecosystem after the fire might have a different heterotrophic respiration e.g. due to decomposition, regrowth, carbon re-allocation, . . . These two datasets should maybe be analysed separately as well as together to give a measure of robustness.

- Generally, the manuscript is missing any uncertainty estimates of the flux calculated from the EC measurements e.g. due to random error, ustar filtering, gap filling, partitioning. . . These are necessary to be able to assess the significance of the results.

- For the regressions, bootstrapping would be useful to give more realistic estimate of the uncertainty than just the standard deviation from the regression.

The following few minor point would favourable for easier reading of the ms

- a table with abbreviations,

- adding the kind of data to the figure descriptions, e.g. "monthly mean" ecosystem respiration in Fig.3,

- explanation of difference between GPP with assimilation A,

- for Section 3.1. adding a figure with the monthly NEE over the six year period.

Hopefully, these comments will help to advance the progress of this ms and I would like to encourage resubmission. By focussing only on the right flow of the partitioning scheme in Figure 1, it would be a short and sweet analysis.

---

## Referee Comment (RC2) · Anonymous Referee #1 · 5 Jul 2016

In my comment on page 1 about the offset C of the light response function, there is part of the sentence missing: "Hence, offset C might be due to the random error in the measurements and would be desired **to be small (close to zero)** as a sign that the partitioning scheme shown in Figure 1 (left flow) worked."

---

## Referee Comment (RC3) · Anonymous Referee #2 · 14 Aug 2016

Koerber et al. propose that they are validating eddy covariance flux measurements with other measurements including leaf area index and ecosystem respiration by using a new – and rather confusing – approach where they adjust the calculated light response curve.

I found the manuscript to be poorly organized and somewhat confusing. I don't agree that the approach helps any overestimate or underestimate of respiration or assimilation if the true value, or at least a value constrained by multiple other observations, isn't known. A visual schematic (more than a flowchart) of the light response adjustments

that they advocate would be helpful; at the moment I can find little justification by the particular approach chosen. A formal uncertainty analysis is certainly needed. The discussion is short and poorly organized, perhaps a bit rushed. Quite a few statements in the Results would make for a compelling Discussion section, although other examples from the literature seem to have been picked almost at random. Altogether an analysis that is important or interesting as we learn that the Kok effect shouldn't be ignored, but ultimately unconvincing. The following minor points are designed to help the authors re-write the manuscript for resubmission.

The statement on line 19 is important but jumped out of the blue a bit. I'm assuming the authors mean the y-intercept of the light response curve? The end of the abstract seems to suggest that, because the numbers are kind of close, the approach must be good. A formal uncertainty analysis is needed. Also, with respect to significant digit reporting, 0.01 g per m2 per year is 10 mg per m2 per year. The eddy covariance method, and all other methods that I know of, are not that certain.

A new manuscript by Wehr et al. (http://www.nature.com/nature/journal/v534/n7609/full/nature17966.html) is critical to cite in any new analysis of daytime respiration suppression.

The first sentence of the intro on line 37 is already a bit of a mouthful. Try to remove every unnecessary word from the manuscript. It could be made much easier to read. (The second sentence beginning line 39 is not a step in the right direction.)

The net ecosystem exchange occurs over the time scale defined, not necessarily a day (line 40).

Keep in mind on line 46 that this statement excludes CAM species.

On line 50, there are (valid) concerns about flux divergence in large-statured forests like tropical forests where the canopy is often warmer than the subcanopy air space, which creates an inversion and suppresses mixing.

On line 52 'decouples' is a word with strong meaning in the atmospheric sciences and

the decoupling is – for lack of a better word – a bit dynamic. I'd reword these passages to note that the decoupling is common.

On what basis is the van Gorsel et al. (2007) approach only valid for undulating terrain?

With respect to the Heskel et al. (2013) paper, please also note the new paper by Wehr et al. (2016). With this in mind, the approach developed by the present manuscript is important. I just wish it was better-written.

There are other reasons why NEE and NEP diverge (noting that the authors did not adequately introduce this discrepancy above). Carbon released by, for example, soil could be taken up by leaves on its way out of the canopy. These fluxes beneath the sensors may not be adequately captured depending on the profiling system, causing further NEE and NEP divergence (I think that Goulden et al. 1996 delves into this issue.)

I think that the approach to use variable LAI is interesting but worry that pre and post-fire ecosystems have rather different heterotrophic respiration rates, making me question the validity of this approach (see line 89).

Hypothesis 1 is a straw-man hypothesis, not a null hypothesis.

Spaces between, for example, 20 and m on line 114 I know will be corrected during the copyrighting process but no harm in starting things off on the right foot.

I like the description of mallee, but because this is a habitat unique to Australia please explain briefly for a global audience.

On what basis was the ustar threshold set to those values for different years? These seem reasonable but there are different ways of computing the threshold, which I personally feel is best-applied seasonally instead of annually. Please justify the choice of the annual ustar threshold.

On line 171what are the 'Kormann-Meixner constraints'?

Surprised in Fig. 1 that 10 W/m2 was chosen as a nighttime threshold. Zenith angle is so much less-prone to uncertainty.

Please explain the soil moisture adjustment on line189 in a bit more detail; the flux response can go both ways in droughted vs. waterlogged conditions.

There is no justification for the approach on line 202 that PAR decreases respiration only above the compensation point.

If nonlinear weighted least squares was used, what was the weighting? It's been argued (e.g. Richardson et al. 2005) that least absolute deviation should be used for parameter fitting.

On line 272 and elsewhere this is far too many significant digits for an eddy covariance measurement of net ecosystem exchange.

Lots of discussion in the results section, i.e. lines 278-281.

There is no justification for saying that the OzFluxQC is underestimating ER on line 322. Or 358 for that matter. (also 404! On what basis do the authors call any of these extimates an over or underestimate when they don't know the true value! This is biased thinking.) The statement on line 325 is also vague.

The statement beginning line 367 makes no sense in context. Why compare these ecosystems only to one in the UK? To add another (self) citation? The same goes for the comparison to Xu et al. 2004. Not incorrect but puzzling given the hundreds of flux studies. The discussion as a consequence is poorly-organize.

The statement on line 426 also makes little sense (much of the manuscript makes little sense). Using nighttime flux observations from eddy covariance is also non-destructive.

Table 2 would be more comprehensible as a figure or two.

[Figure]

---

## Author Comment (AC1) · 10 Sep 2016

Anonymous Referee #1 (AR#1)

The manuscript presents a promising idea for partitioning ecosystem respiration into autotrophic and heterotrophic respiration: Since leaf area index (LAI) is directly related to the autotrophic respiration, the y-intercept from the regression of ecosystem respiration against LAI would be the heterotrophic respiration. Due to a severe bush fire, the ecosystem around the eddy covariance (EC) flux tower was severely damaged and therefore LAI showed a wide variation over the six reported years to test this hypothesis.

However, there seems to be error in reasoning in the light response approach for estimating the ecosystem respiration:
- The net ecosystem exchange (NEE) is first partitioned into average adjusted night EC flux and assimilation A. Hence A is purely the assimilation WITHOUT night time respiration. However, in the light response approach, A is fitted against light with an offset C and for zero light, C is described and discussed as night time respiration R night (which has already been subtracted beforehand). That does not seem to make any sense or needs to be clarified.
- Furthermore, the offset C is quite small for all calculated months (Table 2) with values below the random error typical for EC measurements (cf. Richardson, A.D. et al., 2008. AgForMet, 148(1): 38-50.).

"Hence, offset C might be due to the random error in the measurements and would be desired to be small (close to zero) as a sign that the partitioning scheme shown in Figure 1 (left flow) worked."

This review process has been very challenging for me to grapple with the terminology, a view I am assured is felt by everyone in the eddy covariance community. I have found great solace and affinity in the opinion piece by Wohlfahrt and Gu (2015) in Plant, Cell and Environment, 38, 2500-2507. I can see now that my method calculates 'true photosynthesis', $V_c$.

I feel very grateful to AR#1 for advancing my knowledge and allowing for my understanding to meet his/her.

What I aimed for my manuscript was to calculate an accurate ecosystem respiration ($R_{eco}$) requiring an accurate gross primary productivity where photorespiration accompanies carboxylation. Unfortunately my method eliminates photorespiration preventing me from calculating an accurate $R_{eco}$. My ultimate aim with an accurate $R_{eco}$, was to partition into autotrophic and heterotrophic, the later to compare with ground measurements to ground truth the eddy covariance data from the tower.

I agree with AR#1 and am able to clarify my light response approach does not deliver night time respiration.

- Some of the monthly regressions in Table 2 yielded ecophysiological implausible values (such as negative light compensation points) and other so low $r^2$ that it is questionable if the light response can be fitted for these months at all (since light does not seem to be the primary driver of the ecosystem response).
- Since the main result of the LAI regression again ecosystem respiration in Figure 4 is derived from monthly ER as the difference between monthly NEE and monthly A (hence no light response function used, right flow in Figure 1), it might be recommendable to totally exclude the light response approach from this manuscript.

As it stands, the AR#1 is correct. My calculations give $V_c$. I admire Wohlfahrt and Gu (2015) for remaining with Brooks and Farquhar (1985) in Planta 165, 397-406 as I endeavour to partition the

eddy covariance flux from a processes point of view and agree wholeheartedly it is imperative to determine GPP or the apparent photosynthesis where it integrates photorespiration ($V_o$). To go over my calculations in order to explain how I plan to improve them; my calculations had the advantage of removing soil respiration ($R_{non-leaf}$) as:

$NEP_d - NEP_n$ when equilibrated for the soil temperature and water content gives;

$V_c - 0.5V_o - (R_{day} + R_{non-leaf}) + - (R_{dark} + R_{non-leaf})$

Delivers: $V_c - 0.5V_o + R_{gap}$

$R_{gap}$ is the amount suppressed by light of the non-photo-respiratory respiration. It immediately reduces apparent photosynthesis to true photosynthesis along with cancelling out of the $R_{non-leaf}$.

In my revised manuscript, I will instead use diurnal traces of soil respiration and instead add the night onto the day such that;

$NEP_d + NEP_n$ when equilibrated for the soil temperature and water content gives;

$V_c - 0.5V_o - (R_{day} + R_{non-leaf}) + - (R_{dark} + R_{non-leaf})$

Delivers: $V_c - 0.5V_o - R_{gap} - 2R_{non-leaf}$

Subtracting double the soil respiration will deliver $V_c - 0.5V_o - R_{gap}$, an estimate of GPP, that will be somewhere between apparent photosynthesis ($V_c - 0.5V_o$) and net photosynthesis ($V_c - 0.5V_o - R_{day}$).

I will feel satisfied using this estimate of GPP to calculate ER for regression against LAI, leading into the next points below.

The other major concern is the robustness of the main results in Figure 3 and in Figure 4 of "monthly NEE minus monthly A" against LAI:
- To my understanding, the used partitioning scheme is the right flow of Figure 1 and does not use the light response equation but directly calculates monthly ER from monthly NEE minus monthly A. However, it is often referred to "light response function of calculated assimilation" e.g. in Figure 3. This would need to be clarified or revised throughout the ms.

The revised manuscript will follow the right flow of Figure 1 with an estimate of GPP as described above.

- It is interesting that the ecosystem respiration estimates from partitioning with OzFlux are so much lower, only 16%, than the estimates found with the new partitioning scheme after Figure 1, right flow. This would definitely require further investigation and discussion.

We can understand now why so different as it is just a carboxylation rate when subtracted from NEE give a much larger $R_{eco}$ than from OzFlux. I can't wait to see the next $R_{eco}$ from a better GPP.

- The main result in Figure 4 show monthly data points for prefire and postfire. The two clouds of data points have very different properties in terms of scatter as well the slope/intercept. As a quick test (by reading the values from the figure), a linear regression of only the postfire data yielded a y-intercept of 0.23 and a very low r2 value of 0.26. The y-intercept is thus only half of the regression of the postfire data points. The disturbed ecosystem after the fire might have a

different heterotrophic respiration e.g. due to decomposition, regrowth, carbon re-allocation. These two datasets should maybe be analysed separately as well as together to give a measure of robustness.

I agree to carry out regressions pre and post fire separately and together.

- Generally, the manuscript is missing any uncertainty estimates of the flux calculated from the EC measurements e.g. due to random error, ustar filtering, gap filling, partitioning. These are necessary to be able to assess the significance of the results.

I have now familiarised myself with uncertainty analysis, accounting for random and systematic errors (Moncrieff et al., 1996) and will follow Hollinger and Richardson (2005) with guidance from Peter Isaac (OzFlux). The 95% confidence intervals will be derived from the Bootstrap method, and I would like to incorporate a table of descriptive statistics for the heat, water vapor and $CO_2$ fluxes ($H$, $LE$ and $F_c$).

- For the regressions, bootstrapping would be useful to give more realistic estimate of the uncertainty than just the standard deviation from the regression.

The following few minor point would favourable for easier reading of the ms
- a table with abbreviations,

A table of abbreviations will be added.

- adding the kind of data to the figure descriptions, e.g. "monthly mean" ecosystem respiration in Fig.3,

Figure descriptions will be added.

- explanation of difference between GPP with assimilation A,

The revised manuscript will use an estimate of GPP.

- for Section 3.1. adding a figure with the monthly NEE over the six year period.

The revised manuscript will add a figure of monthly NEE over the six year period.

Hopefully, these comments will help to advance the progress of this ms and I would like to encourage resubmission. By focussing only on the right flow of the partitioning scheme in Figure 1, it would be a short and sweet analysis.

---

## Author Comment (AC2) · 10 Sep 2016

Koerber et al. propose that they are validating eddy covariance flux measurements with other measurements including leaf area index and ecosystem respiration by using a new – and rather confusing – approach where they adjust the calculated light response curve.

I found the manuscript to be poorly organized and somewhat confusing. I don't agree that the approach helps any overestimate or underestimate of respiration or assimilation if the true value, or at least a value constrained by multiple other observations, isn't known. A visual schematic (more than a flowchart) of the light response adjustments that they advocate would be helpful; at the moment I can find little justification by the particular approach chosen. A formal uncertainty analysis is certainly needed. The discussion is short and poorly organized, perhaps a bit rushed. Quite a few statements in the Results would make for a compelling Discussion section, although other examples from the literature seem to have been picked almost at random. Altogether an analysis that is important or interesting as we learn that the Kok effect shouldn't be ignored, but ultimately unconvincing. The following minor points are designed to help the authors re-write the manuscript for resubmission.

The statement on line 19 is important but jumped out of the blue a bit. I'm assuming the authors mean the y-intercept of the light response curve?

The revised manuscript will be better organised because my understanding is now complete. The revised manuscript will walk readers though to my ultimate aim of calculating a correct ecosystem respiration $R_{eco}$ for regressing against leaf area index (LAI). The y-intercept will be the heterotrophic soil respiration ($R_h$). Hence if I can derive a correct $R_{eco}$ then we can use LAI to partition $R_{eco}$ into heterotrophic and autotrophic respiration.

The end of the abstract seems to suggest that, because the numbers are kind of close, the approach must be good. A formal uncertainty analysis is needed. Also, with respect to significant digit reporting, 0.01 g per m$^2$ per year is 10 mg per m$^2$ per year. The eddy covariance method, and all other methods that I know of, are not that certain.
A new manuscript by Wehr et al.

(http://www.nature.com/nature/journal/v534/n7609/full/nature17966.html) is critical to cite in any new analysis of daytime respiration suppression.

The revised manuscript will contain a formal uncertainity analysis. Thank you for drawing my attention to the new paper by Wehr et al., 2016.

The first sentence of the intro on line 37 is already a bit of a mouthful. Try to remove every unnecessary word from the manuscript. It could be made much easier to read. (The second sentence beginning line 39 is not a step in the right direction.)

The revised manuscript will pay better attention to removing unnecessary words.

The net ecosystem exchange occurs over the time scale defined, not necessarily a day (line 40). Keep in mind on line 46 that this statement excludes CAM species.

Thank you for making sure the NEE timescale will be correctly defined and that we are not considering CAM species at night is an important oversight that will be stated in the revised manuscript.

On line 50, there are (valid) concerns about flux divergence in large-statured forests like tropical forests where the canopy is often warmer than the sub-canopy air space, which creates an inversion and suppresses mixing. On line 52 'decouples' is a word with strong meaning in the atmospheric sciences and the decoupling is – for lack of a better word – a bit dynamic. I'd reword these passages to note that the decoupling is common.

We will reword decoupling as you suggest in the revised manuscript.

On what basis is the van Gorsel et al. (2007) approach only valid for undulating terrain?

By this statement we meant we don't have to use the van Gorsel et al. (2007) approach because our site is not undulating. In the revised manuscript we will clarify.

With respect to the Heskel et al. (2013) paper, please also note the new paper by Wehr et al. (2016). With this in mind, the approach developed by the present manuscript is important. I just wish it was better-written.

I understand and I thank both of the reviewers for their constructive comments. My knowledge and understanding of eddy covariance, photosynthesis and carbon budgeting is now complete and the revised manuscript will be better-written.

There are other reasons why NEE and NEP diverge (noting that the authors did not adequately introduce this discrepancy above). Carbon released by, for example, soil could be taken up by leaves on its way out of the canopy. These fluxes beneath the sensors may not be adequately captured depending on the profiling system, causing further NEE and NEP divergence (I think that Goulden et al. 1996 delves into this issue.)
I think that the approach to use variable LAI is interesting but worry that pre and post-fire ecosystems have rather different heterotrophic respiration rates, making me question the validity of this approach (see line 89).

In the revised manuscript, pre and postfire will be analysed separately and together.

Hypothesis 1 is a straw-man hypothesis, not a null hypothesis.

Thank you, seen as hypothesis 1 is self evident, it will be removed.

Spaces between, for example, 20 and m on line 114 I know will be corrected during the copyrighting process but no harm in starting things off on the right foot. I like the description of mallee, but because this is a habitat unique to Australia please explain briefly for a global audience.

The revised manuscript will describe mallee in a global context.

On what basis was the ustar threshold set to those values for different years? These seem reasonable but there are different ways of computing the threshold, which I personally feel is best applied seasonally instead of annually. Please justify the choice of the annual ustar threshold.

The OzFlux processing calculates ustar on an annual basis. I will incorporate seasonal ustar in the revised manuscript if possible.

On line 171what are the 'Kormann-Meixner constraints'?

The revised manuscript will now contain a better description like Kljun et al, 2003 as follows:

To calculate the effective sampling footprint of the tower we used the Kormann-Meixner method (Kormann and Meixner, 2001), employing a modified version of the ART Footprint Tool of Neftel et al. (2008). The Kormann-Meixner footprint determines the two-dimensional density function for an ellipse upwind from the tower. "The model is based on the assumption of a homogeneous and stationary flow over a homogeneous terrain. It can be applied to a large range of atmospheric stability conditions, stable and unstable. By assuming an independent vertical and crosswind dispersion the continuity equation is reduced to a two dimensional advection-diffusion equation, which are solved utilising power law profiles for the horizontal wind velocity and the eddy diffusivity. The predominant wind direction here is from the south-westerly quarter. For every 30 minute measurement of wind speed and direction, mixing and buoyancy parameters a "footprint" is calculated. Analysis of the seasonal effects exhibited a smaller footprint in summer which reflected the increased mixing in summer as well as the influence of more frequent winds from the northerly quarter. The annual average of the footprint area for 2014 displayed a distance from the tower of 500 m for at least 10% of the maximum contribution (1300 m for at least 1%).

Surprised in Fig. 1 that 10 $W/m^2$ was chosen as a nighttime threshold. Zenith angle is so much less-prone to uncertainty. Please explain the soil moisture adjustment on line189 in a bit more detail; the flux response can go both ways in droughted vs. waterlogged conditions.

Thank you, soil moisture will be described in more detail and you are correct the adjustment can go both ways.

There is no justification for the approach on line 202 that PAR decreases respiration only above the compensation point.

Thank you, I will correct this incorrect part of the sentence.

If nonlinear weighted least squares was used, what was the weighting? It's been argued (e.g. Richardson et al. 2005) that least absolute deviation should be used for parameter fitting.

We will take the same approach as Stoy et al., (2006), where they noted that the error distribution of EC measurements may be better approximated as double exponential (Laplacian), rather than normal (Gaussian), and thus least absolute regression maybe preferred for estimating model parameters. For the same reasons as Stoy et al., (2006) we employed least-squares optimization here for comparisons with previous studies.

Furthermore, some months had PAR thresholds less than 1500 $\mu mol\ m^{-2}$ and therefore the majority of our data will be in the bottom of a Gaussian distribution not requiring a Laplacian, peaked distribution.

On line 272 and elsewhere this is far too many significant digits for an eddy covariance measurement of net ecosystem exchange.

Thank you, the revised manuscript will carry out calculations correctly retaining the initial decimal places.

Lots of discussion in the results section, i.e. lines 278-281.

Lines 278-281 will be saved for the discussion.

There is no justification for saying that the OzFluxQC is underestimating ER on line 322. Or 358 for that matter. (also 404! On what basis do the authors call any of these estimates an over or underestimate when they don't know the true value! This is biased thinking.) The statement on line 325 is also vague.

Researchers measuring leaf respiration directly always refer to not incorporating the "Kok" effect will lead to overestimates of ecosystem respiration and gross primary productivity, see Heskel et al., (2013). I can understand AR#2's concerns and will clarify in the revised manuscript.

The statement beginning line 367 makes no sense in context. Why compare these ecosystems only to one in the UK? To add another (self) citation? The same goes for the comparison to Xu et al. 2004. Not incorrect but puzzling given the hundreds of flux studies. The discussion as a consequence is poorly-organize.

I will add more references besides mine.

The statement on line 426 also makes little sense (much of the manuscript makes little sense). Using nighttime flux observations from eddy covariance is also non-destructive. Table 2 would be more comprehensible as a figure or two.

Thank you, you are correct and I will remove this.

---

## Editor Comment (EC1) · M. Reichstein (Editor) · 15 Sep 2016

Thanks for the overall constructive discussion. Just one point regarding the weighted least squares (La Place or Gaussian). Please consider "Lasslop, G., Reichstein, M., Kattge, J., Papale, D., 2008. Influences of observation errors in eddy flux data on inverse model parameter estimation. Biogeosciences 5, 1311-1324.", who show that the LaPlace distribution can be apparent only and caused by heteroscedasticity. Based on your assessment, you ought to choose absolute errors or squared errors (or try both and check how robust the results are).